# Prevalence and Toxicity Characterization of *Bacillus cereus* in Food Products from Poland

**DOI:** 10.3390/foods8070269

**Published:** 2019-07-19

**Authors:** Anna Berthold-Pluta, Antoni Pluta, Monika Garbowska, Ilona Stefańska

**Affiliations:** 1Division of Milk Biotechnology, Department of Biotechnology, Microbiology and Food Evaluation, Faculty of Food Sciences, Warsaw University of Life Sciences—SGGW, Nowoursynowska 159C St, 02-787 Warsaw, Poland; 2Department of Preclinical Sciences, Faculty of Veterinary Medicine, Warsaw University of Life Sciences—SGGW, Ciszewskiego 8 St, 02-787 Warsaw, Poland

**Keywords:** *Bacillus cereus*, food products, Hbl toxin, Nhe toxin, *ces* gene

## Abstract

The prevalence of *Bacillus cereus* in a total of 585 samples of food products (herbs and spices, breakfast cereals, pasta, rice, infant formulas, pasteurized milk, fresh acid and acid/rennet cheeses, mold cheeses and ripening rennet cheeses) marketed in Poland was investigated. The potential of 1022 selected isolates of *B. cereus* to hydrolyze casein, starch and tributyrin, to ferment lactose, to grow at 7 °C/10 days, to produce Nhe and Hbl toxin and to possess the *ces* gene was verified. *B. cereus* was found in 38.8% of the analyzed samples, reaching levels from 0.3 to 3.8 log CFU g^−1^ or mL^−1^. From the 1022 isolates, 48.8%, 36.0%, 98.9%, 80.0% and 25.0% were capable of fermenting lactose, producing amylase, protease, lipase and growing at 7 °C/10 days, respectively, indicating spoilage potentiality. The occurrence of toxigenic *B. cereus* strains in all tested market products, both of plant (55.8% Hbl(+), 70.7% Nhe(+) and 1.7% *ces*(+) isolates) and animal origin (84.9% Hbl(+), 82.7% Nhe(+) and 0.9% *ces*(+) isolates) indicates the possible risk of foodborne infections/intoxications that occur as a result of the possibility of the development of *B. cereus* in favorable conditions and consumption of these products.

## 1. Introduction

Natural environmental reservoirs of the spore-forming bacteria of the *Bacillus cereus* species include the soil, plant surfaces and contaminated water [1]. Due to their ubiquitous distribution in the natural environment, these microorganisms constitute part of the permanent microflora in various raw foods and foodstuffs, including cereal grains and cereal products [2,3], milk and dairy products [2,4,5,6,7,8], fruits, vegetables and their preserves [9,10] and also spices [11]. *B. cereus* bacteria were also isolated from commercial ground roasted coffee [12] and ready-to-eat meals and products [13]. 

*B. cereus* bacteria can cause two types of food-poisoning which vary in their pathogenic mechanism and course. The diarrheal syndrome is a toxicoinfection induced by the three pore-forming enterotoxins cytolysin K (CytK), non-hemolytic enterotoxin (Nhe), and hemolysin BL (Hbl) produced by vegetative cells in the small intestine. CytK is a single-protein toxin that was identified during a food poisoning outbreak in France in 1998 [14]. Nhe and Hbl are both tripartite toxins that require the simultaneous action of the three proteins NheA, NheB, and NheC, or Hbl-B, Hbl-L_1_, and Hbl-L_2_, respectively. The ability to form Hbl and Nhe toxins was found in 42–73% and 97–99% of the strains isolated from cases of food poisoning [15]. The diarrheal syndrome of food-poisoning is characterized by abdominal pains and watery diarrhea appearing 8–16 h after the ingestion of food containing vegetative cells or spores of pathogenic *B. cereus* strains. The estimated count potent to induce the diarrheal type of food poisoning ranges from ca. 5 to 8 log CFU of *B. cereus* vegetative cells or spores [16,17]. By contrast, the first symptoms of the emetic type of food poisoning (nausea, vomiting, headaches) occur already within 1–5 h after the consumption of food containing cereulide-heat-and gastric acid-resistant peptide. This type of food poisoning is most often associated with the ingestion of cereal products, especially rice [16,17]. 

The aim of this study was to determine the prevalence of *B. cereus* in different food products, such as herbs and spices, rice, pasta, cereals, infant formulas, milk and cheeses. *B. cereus* isolates were further assayed for their ability to hydrolyze casein, starch and tributyrin, and their ability to ferment lactose, grow at 7 °C, and produce NHE and Hbl toxin. Further, *B. cereus* isolates were checked for the presence of the *ces* gene.

## 2. Materials and Methods

### 2.1. Prevalence of Bacillus cereus in Food Products

#### 2.1.1. Sample Collection

The studies were conducted in 2007–2017. The commercial food products were purchased from retail shops within the territory of Warsaw and subjected to examination before their use by date. They were transported to the laboratory in a cold box (below 4 °C), and analyzed immediately. Altogether, 585 samples of food products were examined. This number included 60 samples of herbs and spices (in 2008 and 2012–2017), 43 samples of breakfast cereals (in 2007 and 2016), 54 samples of pasta (in 2008 and 2016), 48 samples of rice (in 2008 and 2017), 30 samples of infant formulas (in 2009 and 2015), 60 samples of pasteurized milk (in 2007 and 2015), 35 samples of fresh acid cheeses and acid/rennet cheeses (in 2007 and 2014), 80 samples of mold cheeses (in 2007–2015) and 175 samples of ripening rennet cheeses (in 2007–2015). 

#### 2.1.2. Enumeration of *Bacillus cereus*

*B. cereus* counts (the total number of vegetative cells and spores) were assayed in the examined products according to the colony count technique ISO 7932 [18]. The product samples were collected under sterile conditions in the amount of 10.0 g (or 10.0 mL in the case of liquid products) and placed in sterile plastic bags containing 90 mL of a diluent (peptone saline solution or 2% sodium citrate solution in the case of cheeses). The mixtures were homogenized for 10 min in a STOMACHER 80 (BA 7020 type, Seward Ltd., Worthing, UK). In this way, 10^−1^ dilutions were obtained, which were subjected to further dilution [19,20]. To enumerate *Bacillus cereus*, the plates were inoculated by spreading the samples on the surface of the medium developed by Mossel (MYP Agar, MERCK, Warsaw, Poland). When determining the number of *B. cereus* in pasteurized milk, additionally two parallel Petri dishes were inoculated with 1 mL sample for each. Once presumptive *B. cereus* species bacteria were detected, their colonies were isolated for identification and confirmation of their affiliation to this species. Five typical colonies were picked from each plate (large, pink, and surrounded by a turbidity zone) and subjected to further tests. Where there were less than 5 colonies, all were isolated. The examination of the isolates was performed using the ISO 7932 procedure taking into account properties regarded as characteristic of *B. cereus*. The isolates were examined for their ability to ferment mannitol and to decompose lecithin on the MYP medium and for their ability to lyse red blood cells on the medium supplemented with 5% sheep blood (MERCK). The isolates identified and confirmed by such examination were stored frozen (at temperature −24 °C) until further analyses. 

#### 2.1.3. Characterization of *Bacillus cereus* Isolates with Respect to Their Capacity for Casein, Starch and Tributyrin Hydrolysis, Lactose Fermentation, Growth at 7 °C and Enterotoxin Production

Altogether, 1022 isolates were obtained from the products tested, including 167 isolates from herbs and spices samples, 37 from breakfast cereals samples, 88 from pasta samples, 59 from rice samples, 39 from infant formulas samples, 86 from pasteurized milk samples, 14 from fresh acid cheeses samples, 191 from mold cheese samples, and 341 from ripening rennet cheeses samples. All these isolates were confirmed as *B. cereus.* Apart from identity confirmation carried out according to the ISO 7932 procedure, all the isolates were tested for their abilities to: Hydrolyze casein, starch and tributyrin; to ferment lactose; to grow at 7 °C/10 days; and to produce diarrheal toxin Hbl. 

The ability of the examined isolates to hydrolyze casein was tested by streaking cultures onto water agar plates supplemented with 15% skimmed UHT milk and incubating them at 30 °C for 24 h. The appearance of clear zones around the colonies indicated the degradation of casein [21]. 

The ability to hydrolyze starch was determined by streaking cultures of the examined isolate onto nutrient agar pre-dried on Petri plates to which 0.25% soluble starch had been added (MERCK). The plates with the streaked cultures were incubated at 30 °C for 3 days. A drop of Lugol’s iodine solution was placed on the edge of the colonies which have grown over this time. The amylolytic properties (hydrolysis of starch) were confirmed by the yellow color of the reagent [21]. 

Tributyrin hydrolysis was tested on a tributyrin agar medium (MERCK), supplemented with glycerol tributyrate (tributyrin) (MERCK) according to the manufacturer’s instructions. The examined *B. cereus* isolates were streaked on the Tributyrin Agar medium pre-dried on Petri plates. The plates were incubated at 30 °C for 3 days. The clear zones around the bacterial colonies indicated a positive test for tributyrin hydrolysis.

The isolates were also tested for the ability to ferment lactose according to the instructions provided by Claus and Berkeley [21]. In brief, a colony of the examined isolate was collected with a loop to test tubes with a liquid medium containing 2 g (NH_4_)_2_SO_4_, 0.5 g yeast extract, 1 g tryptone, 3.22 g Na_2_HPO_4_, 0.12 g KH_2_PO_4_, 0.17 g phenol red and 5.0 g lactose per liter. The mixtures were incubated at 30 °C for 48 h. The color change of the medium from red to yellow indicated lactose fermentation.

To determine the psychrotrophic properties, the examined isolates of *B. cereus* were suspended with a loop in a Ringer’s solution with turbidity equivalent to 2 McFarland. Next, the colonies were streaked on Tryptone Soya Agar pre-dried on Petri plates (OXOID ARGENTA, Poznań, Poland). The plates were incubated at 7 °C for 10 days. 

The ability to produce Hbl and Nhe toxins was measured by using BCET-RPLA test kit (OXOID ARGENTA) and Tecra BDE VIA™ kit (NOACK Poland Sp. z o.o., Warsaw, Poland). The tests were carried out according to the directions supplied by the manufacturers, however, the colonies were screened only for the presence of the toxin. The BCET-RPLA kit measures the L_2_ component of the Hbl enterotoxin complex and the BDE VIA kit measures the NheA component of the Nhe complex [22]. 

#### 2.1.4. Detection of Emetic Toxin Gene

DNA Extraction

Total genomic DNA was obtained using the Syngen DNA Mini Kit (Syngen Biotech Sp. z o.o., Wrocław, Poland). The amount and quality of DNA was determined using the Thermo Scientific NanoDropTM 1000 Spectrophotometer (Thermo Scientific, Waltham, MA, USA).

Polymerase Chain Reaction (PCR)

The *B. cereus* isolates were analyzed for the presence of the emetic toxin gene using the primers CesF1 and CesR2 [23]. The reaction mixture contained 10 μM of each primer, 12.5 μL of DreamTaq PCR Master Mix (2×) (Fermentas, Thermo-Fisher Scientific Inc., Waltham, MA, USA), 100 ng DNA and water up to 25 μL. The following amplification procedure was used: Initial denaturation at 94 °C for 4 min., 40 cycles of 94 °C for 30 s, 54 °C for 45 s, 72 °C for 1 min and the final extension step at 72 °C for 7 min. The PCR product in a total volume of 15 μL was separated in 1.0% agarose gel stained with Midori Green DNA Stain (Nippon Genetics Europe GmbH, Dueren, Germany). GeneRuler 100 bp Plus DNA Ladder was used for estimating the molecular size weight of the obtained band (Fermentas, Thermo-Fisher Scientific Inc.). The DNA from the reference strains was used as positive control—*B. cereus* NCTC 11143 and negative control—*B. cereus* ATCC 14579.

## 3. Results and Discussion

### 3.1. Prevalence of Bacillus cereus in Food Products

The prevalence of *B. cereus* in various food products was illustrated in Table 1. Among the examined products, *B. cereus* has been most frequently found in herbs and spices. The percentage of positive samples reached 63.3% and *B. cereus* counts in these products ranged from 1.0 to approximately 3.0 log CFU g^−1^. The presence of *B. cereus* was found mainly in samples of dried herbs (bay leaves, herbs de Provence, thyme, oregano, marjoram, parsley leaves, fennel leaves, basil, tarragon, and lovage), while in any sample of allspice, rosemary and coriander. The highest contamination was found in samples of thyme and herbs de Provence (>2.0 log CFU g^−1^). When comparing these results with data from literature, it can be concluded that the percentage of contaminated *B. cereus* samples was comparable, yet the contamination level was significantly lower than that reported in the literature. According to literature data, the prevalence of *B. cereus* in dried herbs and spices was at 13.5–85% and counts ranged from ca. 2.0 to 6.0 log CFU g^−1^ [11,24,25,26,27,28], which is in agreement with our findings.

The contamination of spices and herbs poses a serious hazard to the microbial quality of foods containing them. This applies, in particular, to ready-to-eat food products which are not subjected to further heat treatment. Herbs and spices are the main source of spore-forming bacteria in such food products as soups, cooked or stewed dishes and sauces, which provide good conditions for the growth of these microorganisms, and, in the case of *B. cereus*, for the occurrence of food poisoning in consumers [29].

The results of the analyses of *B. cereus* prevalence in commercial cereal grain food products revealed that *B. cereus* bacteria were present in 41 out of 145 samples (28.3%). *B. cereus* was detected in 18.6% samples of breakfast cereals and its counts ranged from 1.0 to 2.2 log CFU g^−1^ (Table 1). The bacteria were more frequently found in pasta rather than in rice, since the percentage of *B. cereus*-positive samples was 37.0% and 27.1%, respectively. The contamination level of pasta samples varied from 1.0 to 3.1 log CFU g^−1^, whereas in the case of rice, it was a little lower—ranging from 1.0 to 2.1 log CFU per gram. 

As *B. cereus* spores are an integral component of soil microflora, they can easily contaminate grains and, as a consequence, cereal grain products, including flour, grits, rice, cereal flakes, pasta and bread. The frequency of *B. cereus* contamination of rice, cereal flakes and pasta found in this research was low, and yet comparable with the results reported by others. In similar studies, *B. cereus* was detected in 26–100% of rice, 20–50% of pasta and 21–89% of breakfast cereals samples. *B. cereus* counts found in the products analyzed in this study were similar to literature data, i.e., from 0.5 to 4.0 log CFU g^−1^ [3,22,30,31,32]. However, in a recent study conducted in the Netherlands, *B. cereus* counts found in the 0.5 % of examined starchy products exceeded 5 log CFU g^−1^ [2].

Dried milk/cereals products (i.e., products to be ingested by a potentially sensitive group of consumers, such as infants and young children) are known to be contaminated with *B. cereus*, especially with its spores [7,33]. Altogether, 30 samples of dry formulas for infants and young children have been tested, where of 9 (30.0%) were found to contain *B. cereus*. In most of the *B. cereus*-positive samples (88.9%), the pathogen counts varied from ≥1.0 to 2.0 log CFU g^−1^. Only 1 sample reached *B. cereus* bacterial count of 2.0 log CFU g^−1^. The isolation frequency and *B. cereus* contamination level of the infant formula samples tested in this research were similar to those reported by other authors. For milk powder and dried milk products with cereals intended for young children, an incidence of 8–52% has been reported, at the bacterial count ranging from 2.0 to 6.0 log CFU g^−1^ [7,33,34]. 

*B. cereus* bacterial species is considered as one of the main microbiological factors limiting the shelf life of pasteurized milk [35]. At counts above 5.0 log CFU mL^−1^, *B. cereus* bacteria may cause flavor and taste defects of pasteurized milk. At higher *B. cereus* counts, the product shows defects, such as sweet curdling and bitty cream (in non-homogenized milk), that result from high proteolytic activity and lecithinase production [36]. In this research, *B. cereus* bacteria were detected in 30% of pasteurized milk samples and their counts were very low, since they did not exceed 0.9 log CFU mL^−1^ in any sample (average 0.4 log CFU mL^−1^) (Table 1). The prevalence of *B. cereus* in pasteurized milk in our study was comparable with those reported previously, unlike the contamination level which was lower. From 33% to 71% of the samples analyzed in India and 100% in China were reported to contain *B. cereus*, with numbers ranging from 1.0 to 4.0 log CFU mL^−1^ [8,34]. 

*B. cereus* bacteria were detected in 3 (8.6%) of the samples of fresh acid cheeses (Table 1). Only in 1 out of 35 examined fresh acid cheeses *B. cereus* counts reached 2.2 log CFU g^−1^, and in the following 2 *B. cereus*-positive samples the counts ranged from ≥ 1.0 to 2.0 log CFU g^−1^. Slightly higher *B. cereus* occurrence frequency was found in Egyptian acid cheeses (30% *B.cereus*-positive samples) [37]. 

*B. cereus* bacteria were found to be present in 42 (52.5%) and 76 (43.4%) of the samples of mold and ripening rennet cheeses, respectively (Table 1). The prevalence of *B. cereus* in rennet cheeses found in our study was greater than in other studies. Namely, for ripening, soft and hard cheeses, *B. cereus* prevalence was reported to reach 25–50% [34,38]. In none of the cheese samples did *B. cereus* count exceed 4.0 log CFU g^−1^. The presence of the *Bacillus* bacteria type, i.e., mostly of *B. cereus* species, was reported for all Ricotta cheese samples, and their count ranged from 1.0 to 3.1 log CFU g^−1^, which is comparable to the results obtained in this study [39]. In contrast, much higher counts of *B. cereus* bacteria group were detected in Indian commercial cheeses (to 6.0 log CFU g^−1^) [34].

In this research, mold cheeses and ripening rennet cheeses were found to be significantly more contaminated than the acid cheeses. It may be caused by a much higher acidity of fresh acid cheeses (pH 4.3–4.6) than the one of ripening rennet cheeses (pH 5.2–5.7) and a lower amount of protein buffering, such a low pH. *B. cereus* spores which can germinate and the vegetative cells which can grow only within less than the first twenty hours of the rennet cheese production process (until the pressing stage). In the next stages, when the pH of the cheeses drops below 6.0, *B. cereus* vegetative cells have no chance to survive and, let alone grow. In the case of fresh acid cheeses or acid/rennet cheeses production, the lowering of milk/curd pH usually occurs much sooner and to a much lower degree. The inactivation of vegetative *B. cereus* cells in the case of all fermented dairy products can also be attributed to other factors, many of which act synergically, such as substrate competitions with lactic acid bacteria and their production of antimicrobial agents (hydrogen peroxide, nisine, formate, acetate or lactate), and the presence of salt and changes to the oxidation-reduction potential in the production process [36]. 

It has been shown that the genotypes of *B. cereus* strains isolated from raw milk differed from genotypes isolated from the production environment and from dairy products, indicating that additional product contamination occurs through reinfection [40]. The unique ability of *B. cereus* to adhere to various surfaces (stainless steel, synthetic materials) and the formation of biofilms, can lead to problems which are hygienic in nature and economic losses due to spoilage of dairy products. *B. cereus* biofilms can develop especially in parts of the production line, which work partially filled, or in which product residues remain after the production cycle (eg pasteurizer, storage tanks). They can then be a source of repeated product reinfection [40,41].

### 3.2. Characterization of Bacillus cereus Isolates

The phenotypic profiles of isolates are shown in Table 2. From 1022 *B. cereus* strains, 499 (48.8%) strains were positive for lactose fermentation, the main carbohydrate of milk. The percentage of lactose(+) strains was significantly higher in the group of isolates obtained from dairy products (86.0%, 71.4%, 56.5% and 76.8% strains isolated from pasteurized milk, fresh acid, mold and ripening rennet cheeses, respectively) than in the group isolated from herbs and spices (8.4% strains) and cereals products (10.8%, 0.0% and 15.2% strains isolated from breakfast cereals, pasta and rice, respectively). The considerable variation between strains isolated from different sources in the ability to degrade lactose was reported by other authors [35,42]. These findings indicate a selection or adaptation of *B. cereus* strains during milk processing. Moreover, although some of *B. cereus* isolates are unable to ferment lactose, they can grow in milk products upon hydrolysis of milk proteins or by glucose consumption following the fermentation of lactose by competitive microorganisms, for example lactic acid bacteria.

In this study, 36.0% strains were positive for the capability to hydrolyze starch (Table 2). As in the case of lactose fermentation, the differences were found in this capability between isolates of *B. cereus* originating from various products. The percentage of *B. cereus* strains isolated from herbs and spices, breakfast cereals, pasta and rice that were confirmed capable of hydrolyzing starch (82.6%, 72.9%, 59.1% and 64.4%, respectively) was within the range reported by other authors for strains derived from different non-dairy food products [3,43,44]. Starch is one of the main components of cereal products, thus the presence of amylase-positive strains of *B. cereus* can lead to potential spoilage of these products. A significantly lower percentage of amylase-producing strains was determined among the isolates obtained from dairy products: Infant formulas (23.1%), milk (2.3%), mold cheeses (11.0%) and ripening rennet cheeses (23.8%). In the case of the strains isolated from fresh acid cheeses, none of them was capable of starch degradation (Table 2). In previous studies, from 0 to 100% strains isolated from milk and dairy products were amylase positive [34]. One of the traits that distinguishes the homogenous groups of emetic-type strains from the remaining *B. cereus* strains is the inability to hydrolyze starch [45]. Considering the above, the prevalence of the emetic subtype of *B. cereus* in food products analyzed in the present study may be found as high (64%). 

*B. cereus* produces various extracellular proteolytic and lipolytic enzymes, which can be responsible for the deterioration of the organoleptic quality and for the shortening of the stability of products, especially milk products. The presence of protease can lead to bitter flavor and sweet curdling of milk. In turn, lipases produced by *B. cereus* cause defects of dairy products like, e.g., bitty cream (phospholipase activity especially in non-homogenized milk and cream) and also contribute to unpleasant off-flavors (like rancid, butyric, unclean and soapy) [46]. The proteolytic and lipolytic activities were found in 92–100% and 50–100% of strains from dairy products [34], while in only in 33% and 27% of isolates from legume-based fermented foods in India [44]. In the present study (Table 2), 98.9% of the isolates were capable of casein degradation, whereas only 2 isolates originating from infant formulas, 2 from mold cheeses and 3 from rennet cheeses were incapable of casein degradation. The activity towards tributyrin degradation was demonstrated for 80.0% of the analyzed isolates. However, it was the most rarely reported among the strains isolated from infant formulas (59.0% of strains) as well as from herbs and spices (62.9% of strains).

A total of 256 isolates out of 1022 isolates (25.0%) from different tested samples (Table 2), showed visible growth at 7 °C within 10 days and thus fitted the commonly accepted definition of psychrotrophs. The significant differences were, however, observed in this trait between the isolates originating from various food products. This trait was reported the least frequently among the isolates obtained from herbs and spices (6.6%), whereas the highest number of psychrotrophic strains was found among the isolates obtained from breakfast cereals (48.6%). Similar, considerable deviations in the ability to grow at temperatures of 6–7 °C in the case of *B. cereus* strains isolated from various food products were indicated in the ample literature data [31,47]. In this respect, Samapundo et al. [31] found that among 380 strains isolated from food products marketed in Belgium, only 2.6% were capable to grow at 7 °C, but at mild temperature abuse conditions (≤10 °C), the growth ability was confirmed for the majority of the strains (87.9%). In turn, Godič Torkar and Seme, [47] found that 56.7% of food isolates exhibited psychrotrophic capabilities. In their study, the high contribution of psychrotrophic strains among the isolates were obtained from dairy products, cheeses in particular (31.5% out of 546 isolates originating from fresh acid, mold and ripening rennet cheeses). The high percentage of psychrotrophs illustrates that they probably have a more significant impact on keeping the quality of products stored under refrigerated conditions than mesophilic *B. cereus*. 

From the tested 1022 isolates, 766 (74.9%) and 803 (78.6%) were able to produce the Hbl and Nhe toxin (Table 3). Among the strains isolated from plant products, this percentage ranged for Hbl toxin from 45.9 to 57.9% and for Nhe toxin, from 61.1 to 81.4%, depending on the product, whereas in the group of strains originating from milk products, it was higher and ranged from 80.2% to 89.1% (Hbl) and from 75.9 to 87.7% (Nhe), respectively. The percentage of toxin-positive *B. cereus* strains reported in the literature strongly varies as well. For example, Ankolekar et al. [22] showed that the percentage of enterotoxic isolates obtained from rice was at 61.4% (Hbl) and 84.3% (Nhe). Similar differences for Hbl toxin, although at lower levels than in our study, were found for the strains isolated from dairy products, i.e., from 29.8% [7] to 72.0% [39]. A significant percentage of strains capable of producing Hbl and Nhe toxin was demonstrated among the isolates obtained from infant formulas (74.3% and 76.9%, respectively). All *B. cereus* strains isolated by Organji et al. [6] from pasteurized milk and infant formulas were capable of producing the Hbl toxin. Further, the results of the PCR analysis indicate that most *B. cereus* isolates from powdered infant formulas and other milk products are potential toxin producers [33,48]. Due to the gravity of this problem, FAO/WHO Expert Consultations concluded that *B. cereus* was among major pathogens associated with powdered infant formula contamination [49]. Similarly, the strains isolated from plant-origin products and ready-to-eat (RTE) foods carried genes required for the production of haemolytic BL (*hblA*, *hblC* and *hblD*) and non-haemolytic enterotoxin (*nheA*, *nheB* and *nheC*) [31,50,51].

The prevalence of the emetic gene *ces* among all *B. cereus* isolates in our study were 1.2%. The *ces* gene was only detected in isolates from pasta (2.3% of isolates), rice (6.8%), pasteurized milk (1.2%), mold cheeses (0.5%) and ripening rennet cheeses (1.2%) (Table 3), but not in isolates from herbs and spices, cereals, infant formulas, and fresh cheeses. Taking into account the origin of isolates, in the group of plant-origin isolates, the percentage of isolates with the *ces* gene was 1.7%, while in the group of isolates from dairy products it was 0.9%. In the literature, the detection rates of *ces* gene were significantly lower compared to the detection of the Hbl and Nhe complex genes [12,50,51]. 

The adaptation of *B. cereus* strains to grow at low temperatures makes them grow rapidly in cold-stored food products, thereby deteriorating their quality. In the case of the psychrotropic strains producing toxins, it may also pose a potential threat to the health and safety of consumers. Among the 256 isolates of *B. cereus* able to grow at 7 °C, only 4 (1.6%) possessed the emetic gene *ces* which had been isolated from rice (one isolate), pasteurized milk (one isolate) and ripening rennet cheeses (two isolates). 

## 4. Conclusions

The prevalence of *B. cereus* in food products marketed in Poland demonstrated significant differences depending on the product, i.e., from 8.6% in acid cheeses to 63.3% in herbs and spices, but the contamination level of all the analyzed products did not exceed 4.0 log CFU g^−1^ / mL^−1^. The results concerning the biochemical abilities of the analyzed isolates indicate significant differences in the enzymatic activity of *B. cereus* strains from various food products. The relatively high percentage (25%) of psychrotrophs among the isolated strains is alarming, and among them, four isolates possessed the *ces* gene. These strains may find favorable conditions for growth in food products or in dishes made of these products in households and cold-stored for too long which may pose a risk to the health of the consumer. The occurrence of toxigenic *B. cereus* strains in all tested market products of plant (55.8% Hbl(+), 70.7% Nhe(+) and 1.7% *ces*(+) isolates) and animal origin (84.9% Hbl(+), 82.7% Nhe(+) and 0.9% *ces*(+) isolates) indicate the possible risk of foodborne infections/intoxications that occur as a result of the possibility of the development of *B. cereus* in favorable conditions and consumption of these products. 

## Figures and Tables

**Table 1 foods-08-00269-t001:** The prevalence of *Bacillus cereus* in the commercial products tested.

Tested Products	Number of Samples	Number (Percent) of Positive Samples	*Bacillus cereus* Count in Positive Samples (log CFU g^−1^ or mL^−1^)			Contamination Level (log CFU g^−1^ or mL^−1^)				
			Minimum	Maximum	Average ± SD	<1	≥1–2	≥2–3	≥3–4	≥4
Herbs and spices	60	38(63.3)	1.0	3.0	1.3 ± 0.54	22	34	2	2	0
Breakfast cereals	43	8(18.6)	1.0	2.2	1.2 ± 0.41	35	7	1	0	0
Pasta	54	20(37.0)	1.0	3.1	1.5 ± 0.77	34	15	2	3	0
Rice	48	13(27.1)	1.0	2.1	1.3 ± 0.36	35	12	1	0	0
Infant formulas	30	9(30.0)	1.0	2.0	1.2 ± 0.31	21	8	1	0	0
Pasteurized milk	60	18(30.0)	0.3	0.9	0.4 ± 0.24	60	0	0	0	0
Fresh acid cheeses	35	3(8.6)	1.0	2.2	1.5 ± 0.59	32	2	1	0	0
Mold cheeses	80	42(52.5)	2.0	3.3	2.7 ± 0.40	38	1	33	8	0
Ripening rennet cheeses	175	76(43.4)	1.0	3.8	1.4 ± 0.76	99	36	33	7	0
Total	585	227(38.8)	0.3	3.8	-	376	115	74	20	0

CFU, colony forming unit; SD, standard deviation.

**Table 2 foods-08-00269-t002:** The biochemical and physiological properties of tested *Bacillus cereus* isolates from various foodstuffs.

Product	Number of Tested Isolates	Number (%) of Isolates Exhibiting the Property Under Test				
		Lactose Fermentation	Starch Hydrolysis	Casein Degradation	Tributyrin Degradation	Growth at 7 °C/10 Days
Herbs and spices	167	14(8.4)	138(82.6)	167(100.0)	105(62.9)	11(6.6)
Breakfast cereals	37	4(10.8)	27(72.9)	37(100.0)	37(100.0)	18(48.6)
Pasta	88	0(0.0)	52(59.1)	88(100.0)	88(100.0)	31(35.2)
Rice	59	9(15.2)	38(64.4)	59(100.0)	59(100.0)	10(16.9)
Infant formulas	39	18(46.1)	9(23.1)	37(94.9)	23(59.0)	4(10.3)
Pasteurized milk	86	74(86.0)	2(2.3)	86(100.0)	65(75.6)	10(11.6)
Fresh acid cheeses	14	10(71.4)	0(0.0)	14(100.0)	14(100.0)	6(42.9)
Mold cheeses	191	108(56.5)	21(11.0)	189(98.9)	179(93.7)	33(17.3)
Ripening rennet cheeses	341	262(76.8)	81(23.8)	338(99.1)	252(73.9)	133(39.0)
Total	1022	499(48.8)	368(36.0)	1011(98.9)	818(80.0)	256(25.0)

**Table 3 foods-08-00269-t003:** The toxicity of *Bacillus cereus* isolates.

Product	Number of Tested Isolates	Number (%) of Positive *Bacillus cereus* Isolates		
		Hbl (BCET RPLA)	Nhe (BDE VIA)	*ces* Gene
Herbs and spices	167	94(56.3)	102(61.1)	0
Breakfast cereals	37	17(45.9)	27(73.0)	0
Pasta	88	51(57.9)	71(80.7)	2(2.3)
Rice	59	34(57.6)	48(81.4)	4(6.8)
Infant formulas	39	29(74.3)	30(76.9)	0
Pasteurized milk	86	69(80.2)	70(81.4)	1(1.2)
Fresh acid cheeses	14	12(85.7)	11(78.6)	0
Mold cheeses	191	156(81.7)	145(75.9)	1(0.5)
Ripening rennet cheeses	341	304(89.1)	299(87.7)	4(1.2)
Total	1022	766(74.9)	803(78.6)	12(1.2)

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
