# Peer review of "Prevalence and Toxicity Characterization of Bacillus cereus in Food Products from Poland"

_foods, 2019, doi:10.3390/foods8070269_

Round 1

Reviewer 1 Report

The manuscript entitled “Prevalence and toxicity characterization of Bacillus cereus in food products” is a very interesting and relevant article specially for food safety area. However, I would like to do some suggestions:

In the introduction section would be advisable to explain what is Hbl, Nhe and ces, and their relevance on the pathogenicity of B. cereus

In the Material and Methods would be advisable to include the city,  in addition to the commercial distributor and the country. For example, OXOID ARGENTA,Poland.... Oxoid, Poznań. Poland. Similar in Thermo-Fisher Scientific Inc... This information (commercial distributor, city, country) should only be included for the first time cited.ç

Figure 1 would not be necessary to be included. It does not provide much information 

Author Response

Title: Prevalence and toxicity characterization of Bacillus cereus in food products
Journal: Foods, MDPI (Ref: 540653)

Dear Reviewer,

All comments to the text have been made. Below is a list of changes made:

1) In the introduction section would be advisable to explain what is Hbl, Nhe and ces, and their relevance on the pathogenicity of B. cereus

Changes were introduced in the revised manuscript in the section Introduction (all second paragraph in Introduction; line 35-49):

            B. cereus bacteria can cause two types of food-poisoning which vary in their pathogenic mechanism and course. The diarrheal syndrome is a toxicoinfection induced by the three pore-forming enterotoxins cytolysin K (CytK), non-hemolytic enterotoxin (Nhe), and hemolysin BL (Hbl) produced by vegetative cells in the small intestine. CytK is a single-protein toxin that was identified during a food poisoning outbreak in France in 2008 [Lund et al. 2010]. Nhe and Hbl are both tripartite toxins that require the simultaneous action of the three proteins NheA, NheB, and NheC, or Hbl-B, Hbl-L1, and Hbl-L2, respectively. The ability to form Hbl and Nhe toxins was found in 42-73% and 97-99% of strains isolated from cases of food poisoning [Sastalla et al 2013]. The diarrheal syndrome of food-poisoning is characterized by abdominal pains and watery diarrhea appearing 8-16 hours after the ingestion of food containing vegetative cells or spores of pathogenic B. cereus strains. The estimated count potent to induce the diarrheal type of food poisoning ranges from ca. 5 to 8 log CFU of B. cereus vegetative cells or spores [14, 15]. By contrast, the first symptoms of the emetic type of food poisoning (nausea, vomiting, headaches) occur already within 1-5 hours after the consumption of food containing cereulide - heat- and gastric acid-resistant peptide. This type of food poisoning is most often associated with the ingestion of cereal products, especially rice [14, 15].

2) In the Material and Methods would be advisable to include the city,  in addition to the commercial distributor and the country. For example, OXOID ARGENTA,Poland.... Oxoid, Poznań. Poland. Similar in Thermo-Fisher Scientific Inc... This information (commercial distributor, city, country) should only be included for the first time cited.ç

Changes were introduced in the revised manuscript in the section Materials and Methods in lines: 110, 113, 118, 135, 139, 140, 152, 155, 156, 163, 164, 172

3) Figure 1 would not be necessary to be included. It does not provide much information 

Figure 1 was removed

On behalf of all authors, and my own, I would like to thank You for the time and effort spent on preparing the review.

Kind regards,

Anna Berthold-Pluta

Reviewer 2 Report

The paper reported a survey of 10 years of analysis on B. cereus fond in different substrates.

Although I can find some merit, in some part (materials and methods and results and discussion) the manuscript shoudl be improved.

in particular:

title: include from Poland

line 46-47: delete the ISo, this is not materials and methods

2.1.1     could you inclyde how many food per yesr by typology?

2.1.2    Why don'y you enumerate also spores?

2.1.2    How many colonies did you pick for positive sample? How?

2.1.3    Why 7°C if some strains are recorded to grow also at 5°C? 

3.1    In which spices? there was one tyolopogy more contaminated? which? please detail this part

Table 1: how did you enumerate 0.3 Log CFU/g? this is not in accordance with the materials and methods where you indicate that the first dilution is 10-1, the sam efor 0.4, 0.9...

Table 1. include standard deviation close to average

lines 183-202: include a discussion on the risk of the presence of B. cereus in dairies especially to the growth potetial of the m.o in these substrates (find appropriate references)

conclusions: improve

Author Response

Title: Prevalence and toxicity characterization of Bacillus cereus in food products
Journal: Foods, MDPI (Ref: 540653)

Dear Reviewer,

All comments to the text have been made. Below is a list of changes made:

1) title: include from Poland

Changes were introduced in the revised manuscript, new title:

Prevalence and toxicity characterization of Bacillus cereus in food products from Poland

2) line 46-47: delete the ISo, this is not materials and methods

Changes were introduced in the revised manuscript, lines: 61-62

3) 2.1.1     could you inclyde how many food per yesr by typology?

Changes were introduced in the revised manuscript, in lines: 73-77

4) 2.1.2    Why don'y you enumerate also spores?

The study indicates the presence of a total counts of Bacillus cereus (vegetative cells and spores). The authors decided to determination of the total counts of B. cereus, due to the fact that in the case of some of the tested products (pasteurized milk, ripening cheeses) does not matter whether B.c. are present in form of vegetative cells, or spores (taking into account the induction of food poisoning and the negative impact on the quality of the products). Of course we know that some products contain only the spores of B. c. (herbs and spices, pasta etc.), whereas the other products are subjected to heat treatment prior to consumption (rice, pasta etc), so important for the safety of the consumer are only spores.

Changes were introduced in the revised manuscript, in line: 79

5) 2.1.2    How many colonies did you pick for positive sample? How?

Five typical colonies were picked from each plate (large, pink, and surrounded by a turbidity zone) and subjected to further tests. Where there were less than 5 colonies - all were isolated.

Changes were introduced in the revised manuscript, in lines: 88-90

6) 2.1.3    Why 7°C if some strains are recorded to grow also at 5°C? 

In the determination of psychrotrophic properties, the conditions included in the ISO 17410 standard were applied (Horizontal method for the enumeration of psychrotrophic microorganisms by means of the colony-count technique at 6.5ºC), which provides for incubation of plates under conditions of 6.5ºC / 10 days.

7) 3.1    In which spices? there was one tyolopogy more contaminated? which? please detail this part

The presence of B. cereus was found mainly in samples of dried herbs (bay leaves, herbs de Provence, thyme, oregano, marjoram, parsley leaves, fennel leaves, basil, tarragon, and lovage), while in any sample of allspice, rosemary and coriander. The highest contamination was found in samples of thyme and herbs de Provence (> 2.0 log CFU g-1).

Changes were introduced in the revised manuscript, in lines: 155-158

8) Table 1: how did you enumerate 0.3 Log CFU/g? this is not in accordance with the materials and methods where you indicate that the first dilution is 10-1, the sam efor 0.4, 0.9...

When determining the number of B. cereus in pasteurized milk, additionally two parallel Petri dishes were inoculated with 1 ml sample for each.

Changes were introduced in the revised manuscript, in section Materials and methods, in lines: 155-158

9) Table 1. include standard deviation close to average

Changes were introduced in the revised manuscript, in Table 1

10) lines 183-202: include a discussion on the risk of the presence of B. cereus in dairies especially to the growth potetial of the m.o in these substrates (find appropriate references)

Changes were introduced in the revised manuscript, in lines: 239-247 and two new references

It has been shown that the genotypes of B. cereus strains isolated from raw milk differ from genotypes isolated from the production environment and from dairy products, indicating that additional product contamination occurs through reinfection [Shaheen et al 2010]. The unique ability of B. cereus to adhere to various surfaces (stainless steel, synthetic materials) and the formation of biofilms, can lead to problems of hygienic nature and economic losses due to spoilage of dairy products. B. cereus biofilms can develop especially in parts of the production line, which work partially filled, or in which product residues remain after the production cycle (eg pasteurizer, storage tanks). They can then be a source of repeated product reinfection [Shaheen et al 2010, Kumari and Sarkar 2016].

11) conclusions: improve

Changes were introduced in the revised manuscript, in lines: 347-356

On behalf of all authors, and my own, I would like to thank You for the time and effort spent on preparing the review.

Kind regards,

Anna Berthold-Pluta

Round 2

Reviewer 2 Report

 no revisions required